# A Systematic Test of Receptor Binding Kinetics for Ligands in Tumor Necrosis Factor Superfamily by Computational Simulations

**DOI:** 10.3390/ijms21051778

**Published:** 2020-03-05

**Authors:** Zhaoqian Su, Yinghao Wu

**Affiliations:** Department of Systems and Computational Biology, Albert Einstein College of Medicine, 1300 Morris Park Avenue, Bronx, NY 10461, USA; zhaoqian.wu@einsteinmed.org

**Keywords:** tumor necrosis factor superfamily, binding kinetics, computational simulations

## Abstract

Ligands in the tumor necrosis factor (TNF) superfamily are one major class of cytokines that bind to their corresponding receptors in the tumor necrosis factor receptor (TNFR) superfamily and initiate multiple intracellular signaling pathways during inflammation, tissue homeostasis, and cell differentiation. Mutations in the genes that encode TNF ligands or TNFR receptors result in a large variety of diseases. The development of therapeutic treatment for these diseases can be greatly benefitted from the knowledge on binding properties of these ligand–receptor interactions. In order to complement the limitations in the current experimental methods that measure the binding constants of TNF/TNFR interactions, we developed a new simulation strategy to computationally estimate the association and dissociation between a ligand and its receptor. We systematically tested this strategy to a comprehensive dataset that contained structures of diverse complexes between TNF ligands and their corresponding receptors in the TNFR superfamily. We demonstrated that the binding stabilities inferred from our simulation results were compatible with existing experimental data. We further compared the binding kinetics of different TNF/TNFR systems, and explored their potential functional implication. We suggest that the transient binding between ligands and cell surface receptors leads into a dynamic nature of cross-membrane signal transduction, whereas the slow but strong binding of these ligands to the soluble decoy receptors is naturally designed to fulfill their functions as inhibitors of signal activation. Therefore, our computational approach serves as a useful addition to current experimental techniques for the quantitatively comparison of interactions across different members in the TNF and TNFR superfamily. It also provides a mechanistic understanding to the functions of TNF-associated cell signaling pathways.

## 1. Introduction

The binding between ligands in the tumor necrosis factor (TNF) superfamily and their receptors plays a pivotal role in regulating signaling pathways that are closely related to inflammatory responses, tissue homeostasis, and cell differentiation (Figure 1) [1,2,3]. The ligands in the TNF superfamily (TNFSF) form homo-trimeric complexes through their conserved C-terminal TNF homology domains (THD). Each THD domain adopts a tertiary structural arrangement with β-sandwich “jelly-roll” topology. Most members in the TNFR superfamily (TNFRSF) [4], on the other hand, are type-I transmembrane proteins containing tandem repeats of cysteine-rich domains (CRDs) in their extracellular regions [5]. The binding interfaces for almost all receptors in the TNFRSF are located at the inter-protomer grooves between every two adjacent subunits of a trimeric ligand [6]. In spite of this structural similarity, the ligand-receptor binding across different members in TNF and TNFR superfamily shows elaborately designed specificity and selectivity. Moreover, mutations in genes that encode TNF or TNFR proteins can often lead into a broad range of pathological consequences. For instance, mutations that reduce the cleavage of TNF receptor 1 (TNFR1) result in increased TNF signaling and systemic auto-inflammatory disorder. As a result, therapeutic intervention that targets the interactions between TNF ligands and receptors becomes a promising treatment for a large variety of diseases, such as autoimmune lymphoproliferative syndrome, rheumatoid arthritis, and multiple sclerosis [7,8].

The knowledge on the binding kinetics of TNF ligand-receptor interactions can largely benefit the development of drugs that modulate these interactions, and further facilitate our understanding of the principles underlying the TNF-mediated cell signaling. Binding constants of ligand–receptor interactions in the TNF and TNFR superfamily have been experimentally measured using various methods [9,10], including surface plasma resonance (SPR) [11], isothermal titration calorimetry (ITC) [12], homogenous time resolved fluorescence (HRTF) [13], and other cellular binding assays. However, most of these experiments were performed under very different conditions. As a result, values of binding parameters collected from different methods often varied over several orders of magnitude. Moreover, it has been brought to attention that ligand–receptor binding can be affected by a number of molecular and cellular factors, such as binding avidity in trimeric TNF ligands and dynamic assembly of TNF receptors in the absence of ligands [14]. Unfortunately, the impacts of these factors on ligand–receptor binding can neither be effectively avoided nor quantitatively estimated in experiments. Additionally, almost all experimental studies on binding between TNF ligands and receptors only provided data of their binding affinities. Little information about rate constants that characterize their binding kinetics can be found in the literature. These inconsistent and biased studies limited our understanding of association and dissociation mechanisms between proteins in the TNF and TNFR superfamily.

Compared to the experimental measurements, computational approaches such as molecular dynamic (MD) simulation serves as an ideal alternative to test biological systems under the conditions that are currently inaccessible in the laboratory [15,16,17,18,19,20,21,22]. All-atom MD simulations have recently been utilized to understand the molecular mechanisms during the processes of protein–protein association [23], as well as dissociation [24]. Unfortunately, due to the high demands for computational resources, this technique has so far only been successfully applied to a limited number of protein complexes [25,26,27,28,29,30,31,32,33,34]. Therefore, a new computational strategy that combines different coarse-grained simulation methods is proposed in this work to study the kinetics of TNF ligand–receptor binding on a systematic level. We first constructed a dataset that contained diverse complexes between TNF ligands and their corresponding receptors in the TNFR superfamily; structures that are currently available in the protein databank (PDB). On the basis of these structures, Monte Carlo (MC) and Brownian dynamic (BD) algorithms were further employed to simulate the physical processes of association and dissociation between ligands and receptors in the complexes, respectively. Although most ligands and receptors in the dataset formed complexes with similar binding interface, our simulation results show that the differences in both their associations and dissociations were remarkable. The functional insights of these differences were proposed. For instance, we found that some soluble decoy receptors bound to ligands with low association and dissociation rates. The slow dynamics shaped the functions of these decoy receptors as inhibitors of signal activation. In contrast, binding between ligands and cell surface receptors was more transient, which was rooted in the dynamic nature of cross-membrane signal transduction. We further show that in some specific cases, the binding stabilities inferred from our simulation results were compatible with existing experimental data. Therefore, our computational approach could be a useful addition to current experimental techniques for quantitatively comparing the interactions between ligands and receptors across different members in the TNF and TNFR superfamily. In general, this study showcases the potential of computational simulations in offering mechanistic understanding to specific biological problems.

## 2. Results and Discussions

### 2.1. Structural Comparison of TNF Ligand–Receptor Complexes in Different Systems

We constructed a non-redundant structural dataset of TNF ligand–receptor complexes in order to study the dynamics of their interactions. The dataset contained 10 different complex structures in which all the ligands in the complexes belonged to the TNF superfamily, whereas their receptors belonged to the TNFR superfamily. The detailed procedure and criteria of data selection are described in the Model and Methods section. Before the simulations of complex formation and dissociation, we first compared the structures of their interactions between ligands and receptors. In detail, only one receptor was left in the comparison for simplification. On the other hand, ligands retained as trimeric organizations because their receptor binding interfaces are located between two adjacent subunits. During the structural comparison, all the trimeric ligands in the dataset were spatially aligned together by rigid-body superposition, so that we were able to highlight the variations in binding of receptors. As a result, with backbone representation of different colors, the positions of all the receptors in the dataset are plotted in Figure 2 around their ligands, wherein subunits are shown in different gradients of grey with surface representation.

Figure 2 indicates that the structures and binding of receptors can be divided into three groups. Specifically, among the receptors in 10 complexes, 7 of them were closely aligned with each other, as shown in Figure 2a. As listed in the color index on the right side of the figure, these receptors are in the complexes of TNF-related apoptosis-inducing ligand (TRAIL)/death receptor 5 (DR5) (PDB ID 1d0g, red), LTα/TNFR1 (PDB ID 1tnr, orange), TNFA/TNFR2 (PDB ID 3alq, yellow), receptor activator of nuclear factor-κB ligand (RANKL)/receptor activator of nuclear factor-κB (RANK) (PDB ID 3qbq, tan), RANKL/osteoprotegerin (OPG) (PDB ID 3urf, pink), LIGHT/decoy receptor 3 (DcR3) (PDB ID 4j6g, purple), and FASL/DcR3 (PDB ID 4msv, ice blue). Although they all lay at the inter-protomer grooves between the dark grey and light grey subunits of the trimeric ligands, we found that the N-termini of the receptors were aligned relatively better than their C-terminal regions. Moreover, the structural alignment further showed that there were slight variations in the orientations of packing between the receptors and ligands. For instance, comparing with the C-terminal regions of receptors in the LTα/TNFR1 and TNFA/TNFR2 complexes, the C-terminal regions of receptors in the LIGHT/DcR3 and FASL/DcR3 complexes were positioned towards the left, whereas their N-terminal regions were biased towards the right. The energetics origin, the impacts on binding kinetics, and the functional implication of these variations in packing orientations will be discussed in the following sections.

The second group contained a pair of functionally related receptors [35]. In specific, TACI (transmembrane activator and CAML-interactor) and BCMA (B cell maturation antigen) are members in TNFRSF that serve as a key regulator of humoral immunity. Both receptors can bind to ligands in TNFSF, such as APRIL (a proliferation-inducing ligand) and BAFF (B cell activation factor). The interactions between these ligands and receptors modulate the immune responses to pathogens and thus are closely related to autoimmune diseases such as rheumatoid arthritis and multiple sclerosis [36,37]. Interestingly, in contrast to most typical TNFRs that contain multiple copies of CRD, the size of BCMA is much smaller and only has one single CRD. Similarly, TACI is also a relatively short TNFR consisting of two CRDs. More importantly, these two domains share more than 50% sequence identity. Previous experimental evidence showed that a shorter form of TACI with its N-terminal domain removed can still bind to its ligand APRIL and induce cell signaling. The structural comparison of BCMA (violet) and short form of TACI (green) is shown in Figure 2b while binding to their common ligand APRIL. The ligand binding of both receptors showed high structural similarity. Finally, ligand binding of the receptor in the third group was highly different from the first two. The receptor 4-1BB bound to its ligand 4-1BBL, the structure of which forms a disulfide-linked dimeric assembly, unlike all other trimeric ligands (Figure 2c) [38].

### 2.2. Simulating Association and Dissociation of Different Complexes in the Dataset

On the basis of the structural comparison, computational simulations were carried out to model the detailed dynamic process of complex formation between different ligands and receptors in the dataset. We first applied a residue-based kinetic Monte Carlo (kMC) method to all the 10 protein complexes. In each system, the monomeric receptor was separated from its trimeric ligand within different values of distance cutoff *d_c_* (Figure 3a). As introduced in the Model and Methods section, under each distance cutoff, 10^3^ simulation trajectories were generated from different initial conformations. We then counted the probability of finding the encounter complexes among these trajectories (Figure 3b). After systematically scanning the values of *d_c_* from 15 to 25Å, we plotted the relation between the distance cutoff and the probability of complexes formation in Figure 3c for all 10 systems in the dataset. The figure shows that the higher probabilities of association were obtained under smaller values of distance cutoff between all pairs of ligands and receptors in the simulations. The association probabilities dropped as the values of distance cutoff increased, which suggests that if ligands and receptors are initially separated farther from each other, then they are less likely to encounter each other before the end of the given simulation duration.

Moreover, the profiles of association probability for different complexes are highly distinctive from each other. The probabilities in some systems are very high, indicating fast association between ligands and receptors. For instance, given the distance cutoff of 15Å, the probability of forming complex between ligand TRAIL and receptor DR5 (PDB ID 1d0g) was higher than 80%, as shown by the black squares in the figure. For the associations between LTα and TNFR1 (1tnr), as well as between RANKL and RANK (3qbq), the probabilities were higher than 40% under small distance cutoff. On the other hand, the low probabilities in many other systems suggest slow association between ligands and receptors. For examples, the average probabilities of complex formation for TNFA/TNFR2 (3alq), APRIL/TACI (1xu1), APRIL/BCMA (1xu2), LIGHT/DcR3 (4j6g), and FASL/DcR3 (4msv) were lower than 10%. These results indicate that the association rates for different systems in the dataset were highly diverse, although they all belonged to the same superfamily of ligands and the same superfamily of receptors. The comparison of our simulation results with currently available experimental data and their biological insights will be discussed in the following results sections.

In addition to association, we also considered the stability of these ligand–receptor complexes. Specifically, coarse-grained Brownian dynamic simulations were carried out to compare the dissociation processes among all the 10 protein complexes in the dataset. For each system, the native structure of the complex was used as the initial conformation. Following the initial conformation, 10 independent simulation trajectories were carried out. Each trajectory contained 10^6^ simulation steps. The intermolecular interactions formed in the initial native conformation by residues between ligands and receptors gradually broke under a stochastic background in the Brownian dynamic simulations, which led into the final dissociation of the complex. The parameters of these intermolecular interactions between different types of residues were derived from a knowledge-based potential, as introduced in the Model and Methods section. The percentage of native contacts left in the dissociating complex was counted along all simulation trajectories. An intermolecular native contact was defined as an interaction between a residue from the trimeric ligand and the other residue from the monomeric receptor which also exists in the native structure of the complex. The percentage of native contacts was then calculated from the ratio between the left numbers of contacts along dissociation versus the original contact numbers. Figure 4a plotted the variation of this percentage along the simulation steps of one trajectory for the complex between LTα and TNFR1. The initial percentage was 1, as in the native structure, whereas it dropped after the simulation started. This suggests that residues at the binding interface between LTα and TNFR1 lost their contacts. The final percentage at the end of the trajectory was 0, indicating that all the native contacts were broken during simulation. The comparison between initial and final conformations of the complex is highlighted in Figure 4b. The ligand and receptor in the native complex are shown in green and red, whereas their structures in the final conformation are shown in blue and yellow, respectively.

The statistical distributions of native contact percentage remaining after simulations calculated from all 10 trajectories are shown in Figure 4c for all the complexes, wherein PDB IDs are indexed along the *x*-axis. Their distributions are plotted as a box-whisker diagram. Similar to the results of ligand–receptor association, the figure indicates that the dissociations of different complexes in the dataset are highly diverse. The distributions of native contact percentage in some systems are relatively high, for example 1xu1 and 3urf, indicating slow dissociation between ligands and receptors. In contrast, the average native contact percentages after simulation were much lower in some other systems, for instance 1xu2 and 6mkz, suggesting that these complexes are easier to dissociate. We link these simulation results to their possible functional insights and compare them with previous experimental measurements in the following results sections.

### 2.3. Linking the Computational Results with Previous Experimental Data

We compared the relative probabilities of both association and dissociation for all the protein complexes in the dataset. In order to justify the reliability of our simulation results, they need to be further compared with the available experimental measurements. Unlike the kinetic rates of binding, the affinities of binding have been intensively evaluated and well documented for many TNF/TNFR ligand–receptor interactions. Therefore, the calculated probabilities of association and dissociation from our simulations were combined together and converted into a stability factor to qualitatively describe the strength of complex formation. Specifically, for a specific complex, the stability factor was defined as *Prob_Diss_/(1-Prob_Ass_)*, in which *Prob_Diss_* is the averaged percentage of remaining native contact calculated from all 10 trajectories and *Prob_Ass_* is the probability of association derived from all 10^3^ trajectories when distance cutoff equals 15Å. The higher values of *Prob_Diss_* indicate the complex is more difficult to dissociate. Similarly, the higher values of *Prob_Aiss_* indicate the complex is easier to associate. As a result, this stability factor was positively correlated with the binding strength. Among 10 protein complexes in the dataset, 7 of them had corresponding data to binding affinities available in the literature. The stability factors were calculated for these systems. Their comparisons with the logarithm of experimentally measured dissociation constants are plotted in Figure 5.

As shown by different symbols in the figure, we further divided the seven protein complexes into different subgroups so that they can be compared within specific biological context. The complexes of LTα/TNFR1 (1tnr) and TNFA/TNFR2 (3alq) consisted of the first group. The binding affinity between LTα and TNFR1 was measured by enzyme-linked immunosorbent assay (ELISA) and its measured dissociation constant was equal to 0.064 nM [39]. On the other hand, the binding affinity between TNFA and TNFR2 was measured by cellular binding study (CBS) and its measured dissociation constant was equal to 0.42 nM [40]. Therefore, the experimental data indicated that binding of LTα/TNFR1 is stronger than TNFA/TNFR2. Our simulation results show that the association probability of LTα/TNFR1 is higher than TNFA/TNFR2 (Figure 3c), whereas the dissociation of LTα/TNFR1 is slower than TNFA/TNFR2 (Figure 4c). As a result, the calculated stability factors for LTα/TNFR1 and TNFA/TNFR2 are consistent with the experimental information, as indicated by the red circles in Figure 5. Additionally, the binding between 4-1BBL and 4-1BB (6mkz) has also been previously measured, whose dissociation constant was equal to 1.86 nM [9]. This suggests that the interaction of 4-1BBL/4-1BB is weaker than LTα/TNFR1 and is higher than TNFA/TNFR2. This result is also compatible with our calculation, as we compared the stability factor of 4-1BBL/4-1BB (yellow diamond) with the red circles in Figure 5.

The second pair of comparison is between the complexes APRIL/TACI (1xu1) and APRIL/BCMA (1xu2). Although the TNF ligand APRIL can bind to both receptors TACI and BCMA, our simulations suggest that binding of APRIL/TACI is much stronger than APRIL/BCMA, More specifically, we found that although both associations of APRIL/TACI and APRIL/BCMA were comparatively slow (Figure 3c), the dissociation of complex between APRIL and BCMA was much faster than the complex between APRIL and TACI (Figure 4c). This result was supported by the experimental evidence. Previous cell-based binding studies measured the affinities for the ligand–receptor interactions of these two systems. The dissociation constant of APRIL/TACI interaction was equal to 0.236 nM, whereas the dissociation constant of APRIL/BCMA interaction was equal to 0.468 nM [9]. The comparison between the experimental data with our calculated values of stability factor is shown by the green triangles in Figure 5. Finally, we also compared the binding between the complexes LIGHT/DcR3 (4j6g) and FASL/DcR3 (4msv). Experimental data indicated that binding of LIGHT/DcR3 is stronger that FASL/DcR3. The measured dissociation constant for LIGHT/DcR3 and FASL/DcR3 was equal to 0.6 nM [41] and 0.8 nM [42], respectively. Consistent with the experimental measurements, our simulations also suggested that the association between LIGHT and DcR3 is relatively faster and their dissociation is relatively slower. The comparison between the experimental data with our calculated values of stability factor is shown by the blue hexagons for these two systems in Figure 5.

### 2.4. The Implication of Our Simulations in the Biological Functions of the TNF Superfamily

Our simulations were not only compatible with experiments, but can also further provide insights to the biological functions of different ligand–receptor interactions. One typical example is the difference in binding between the ligand, called receptor activator of nuclear factor-κB ligand (RANKL), and its receptor, known as receptor activator of nuclear factor-κB (RANK), from the binding to its competitor osteoprotegerin (OPG). It is well understood that the RANKL–RANK–OPG regulatory pathway plays an important role in controlling osteoclastic bone resorption [43]. RANKL is expressed by osteoblast lineage cells and later binds to RANK, which is a cell surface receptor on pre-osteoclasts and mature osteoclasts. The binding leads to increased bone resorption. On the other hand, OPG is also produced by osteoblasts. It is a soluble “decoy receptor” that also binds to RANKL and inhibits osteoclastic bone resorption by preventing RANKL from binding to RANK (Figure 6a). As a result, the balance of RANKL and OPG released from the osteoblast ultimately determines the rate of bone resorption.

Comparing our simulations of RANKL/RANK (3qbq) with RANKL/OPG (3urf), we found that the association between RANKL and RANK was faster than the association between RANKL and OPG (Figure 3c), whereas the dissociation of RANKL/RANK was also faster than the RANKL/OPG complex (Figure 4c). This result suggests that the binding between RANKL and RANK is more dynamic. This dynamic feature of binding between RANKL and RANK is well compatible with the function of RANKL as an activator of bone remodeling pathways. Through fast association and dissociation with the receptor RANK on the surface of osteoclasts, the initiation or termination of intracellular signal transduction can be processed promptly, thereby maximizing the fidelity of information transferred for osteoclastic bone resorption. On the other hand, the association between RANKL and OPG cannot be too fast, otherwise they would form complexes before RANKL reached the surface of osteoclasts, thereby losing the function of OPG as an inhibitor of RANKL/RANK binding. However, the slow dissociation after RANKL and OPG form complexes can effectively prevent the further binding of RANKL to RANK, therefore maintaining the balance of RANKL/RANK binding and the rate of bone resorption.

We further found that the slow dynamics of binding between ligand and decoy receptor is not limited to RANKL/OPG complex. The binding of another soluble decoy receptor, decoy receptor 3 (DcR3) [44], to two of its ligands, LIGHT and FASL, also showed similar behavior. The calculated probabilities of association for LIGHT/DcR3 and FASL/DcR3 complexes were among the lowest, compared with others in the dataset (Figure 3c). In contrast, the average values of native contact percentage remaining after dissociation simulations for these two systems were among the highest (Figure 4c). In order to explore the energetic origin of this phenomenon, we compared the ligand-binding structure of DcR3 with death receptor 5 (DR5) (Figure 6b). Our study showed that both association and dissociation between DR5 and its ligand TNF-related apoptosis-inducing ligand (TRAIL) were fast. This is opposite to the binding of decoy receptors. The structural comparison revealed the difference of binding between DcR3 and DR5 to their ligands. In detail, a large portion of binding interface in the complex of FASL/DcR3 (PDB ID 4msv) is involved in the backbone hydrogen bonds between DcR3 and the DE loop in the ligand, as shown in Figure 6c. These extended hydrophobic interactions result in the slow association and dissociation. Additionally, these sequence-independent interactions also explain why DcR3 can bind to different ligands with relatively low selectivity [45,46,47]. In contrast, these backbone hydrogen bonds were not presented in the ligand binding of DR5. Alternatively, electrostatic interactions between the polar residues at the interface of TRAIL/DR5 complex (PDB id 1d0g) were observed, as shown in Figure 6d. These interactions between DR5 and its ligand are the driven force of its fast binding kinetics, as well as the basis of high binding specificity.

## 3. Conclusions

Inflammation is a complicated response of the innate immune system to the invasion of foreign pathogens [48]. At the onset of inflammation, chemical factors, called cytokines, are released from injured cells as signals to recruit leukocytes, mainly neutrophils, so that they can remove all the pathogens by phagocytosis. TNF is one major class of cytokines in inflammation that is originally expressed on cell surfaces and later oligomerized into soluble trimers after being cleaved by metalloproteases (Figure 1) [3]. The binding of proteins in the TNF superfamily with their corresponding receptors in the TNFR superfamily initiates multiple intracellular signaling pathways, including the NF-κB signaling pathway as a critical regulator of leukocyte activation (Figure 1) [49,50,51,52]. Therefore, the study of ligand–receptor interactions in the TNF and TNFR superfamily, especially to measure the kinetic rates of their binding, is a significant step to understand the molecular mechanism of cell signaling in the innate immune system. Unfortunately, most of the current available data measuring TNF/TNFR binding were performed by various experimental techniques and under various conditions, leading into the fact that the binding data that were collected from different studies about the same molecular system could differ over several orders of magnitude. Moreover, relative to the studies of binding affinities, very few data can be found in the literature to characterize the kinetics of binding between TNF ligands and receptors.

In order to overcome these limitations, we developed a computational strategy to complement current experimental studies. A kinetic Monte Carlo algorithm that was specifically designed to simulate how fast two proteins form a complex and a coarse-grained Brownian dynamic algorithm that was specifically designed to simulate how fast two proteins in a complex separate were applied to compare the association and dissociation of different TNF ligand–receptor systems, respectively. To test these methods, we selected 10 representative complexes from the protein databank as a non-redundant structural dataset that carries diverse functions in the TNF and TNFR superfamily. For the systems where binding affinities have been measured, we compared our computational results with the experimental data. Although our simulations can only provide relative insights to the binding stability of ligand–receptor interactions, the results derived from our computational calculations and previous experimental measurements were still consistent with each other. We further found that, different from the binding between ligands and cell surface receptors, which show relatively fast association and dissociation in carrying out their dynamic functions in cross-membrane signal transduction, some soluble decoy receptors bind to ligands with both low association and dissociation rates. We showed that this change of binding kinetics between decoy and cell surface receptors originates from the subtle difference in the structures at the binding interfaces between these receptors and their ligands. We suggest that the slow binding dynamics of these decoy receptors are naturally designed to fulfill their functions as inhibitors of signal activation. In summary, we demonstrated that our method could play an important role in providing mechanistic understanding to the function of TNF-mediated cell signaling in addition to current experimental approaches.

TNF receptors are membrane proteins anchored on cell surfaces. Intuitively, binding of these membrane receptors to their ligands has very different properties from interactions of proteins in solution [53]. One major reason is that the diffusions of TNFR are confined on cell surfaces. Unlike freely diffusive soluble proteins, which possess of three translational and rotational degrees of freedom, membrane-anchored proteins not only lose one translational degree of freedom that is perpendicular to the membrane surface, but also obtain constraints on their rotational degrees of freedom [54]. Moreover, TNF ligands generally organize into a homo-trimeric quaternary structure before cleaved by metalloproteases and exposure to TNFRs. This leads into the assembly of a minimal ligand–receptor functional unit with a 3:3 stoichiometric combination [6]. It has been further found that, for different members in TNFRSF, the formation of these complexes can trigger their spatial oligomerization into higher-order clusters and initiate intracellular signaling pathways [55]. These dynamic processes can be simulated by a lower-resolution computational model recently developed by our lab [56]. However, before the study of collective behaviors, which involves multiple ligands and receptors in their cellular environments, one needs to estimate the kinetic properties of binding between an individual TNF and its monomeric receptor as a basis. The computational mothed in this paper can isolate the effect of ligand–receptor binding from a number of above-mentioned complexities at the cellular level. As a result, we are able to only focus on the energetic and structural features in binding of monomeric TNF receptor to the ligand in which the interface is located at groove regions between every two adjacent subunits of a trimer. In the future, the association and dissociation rates derived from this study for an individual pair of ligand–receptor interactions can then be fed into our lower-resolution simulation method to understand the mechanism of receptor clustering on cell surfaces as a multiscale modeling framework.

## 4. Model and Methods

### 4.1. Construction of a Structural Dataset for TNF Ligand–Receptor Complexes

In order to construct a dataset for systematically testing the kinetics of TNF ligand–receptor complex formation, we went through the protein databank (PDB) and found all the structures of protein complexes in which their ligands belong to the TNF superfamily and the receptors belong to the TNFR superfamily. As a result, a total number of 21 records could be found in the database. A subset of complexes among all these records were then selected as the final testing dataset on the basis of the following criteria. Firstly, the redundancy in the dataset was avoided. For instance, if multiple structures of the same ligand–receptor complexes were found, only one was selected as the representative. Similarly, if one ligand–receptor complex was in the dataset, the structures of its homolog from other different organisms would be excluded. Moreover, any complex that contained mutated residues either in its receptor or in its ligand was not included in the dataset, considering that the mutations can affect the kinetic properties of binding between ligands and receptors. Secondly, the biologically diversity of members in the dataset was also taken into account. On the other hand, the functionally related systems needed to be presented together, so that they could be quantitatively compared with one another. On the basis of these criteria, a total number of 10 ligand–receptor complexes were finally selected as the dataset for our study. The detailed information about these complexes is listed in Table 1.

The biological functions of some complexes in the dataset were closely related with each other. For an example, ligand RANKL can bind to both receptors RANK and OPG to regulate signaling in bone resorption. Therefore, we included both complexes of RANKL/RANK (PDB ID 3qbq) and RANKL/OPG (PDB ID 3urf) in our dataset. Furthermore, the proliferation of primary B and T cells can be stimulated by a ligand APRIL. APRIL carries out its functions to B cells via binding to a receptor called BCMA and to T cells via binding to a receptor called TACI. As a result, both complexes of APRIL/BCMA (PDB ID 1xu1) and APRIL/TACI (PDB ID 1xu2) were also included in our dataset. Finally, the decoy receptor DcR3 neutralizes the biological functions of three members of ligands in the TNF superfamily, two of which are included in our dataset. They are complexes of FASL/DcR3 (PDB ID 4msv) and LIGHT/DcR3 (PDB ID 4j6g). The comparison of binding dynamics between these functionally related ligand–receptor interactions across different members in the TNF and TNR superfamily, as well as between their experimental measurements and our simulation results, can help us understanding the molecular mechanism of TNF-mediated cell signaling. More detailed comparisons of these systems from our simulation results were discussed in the Results section.

### 4.2. The Kinetic Monte Carlo Simulations of Association between Ligands and Receptors

A previously developed kinetic Monte Carlo algorithm [27] was applied to simulate the association between ligands and receptors in the TNF and TNFR superfamily. This method was tested against a large benchmark set and showed strong correlation with experimental measured rates of association. In detail, a coarse-grained model of protein structures was used in the simulation. Each residue in the model was represented by two points: one was the Cα atom and the other was the representative center of its side-chain. The side-chain representative centers of different residues were selected on the basis of their specific properties. The simulation started from an initial conformation, in which the structural models of a monomeric receptor and a trimeric ligand were separated away from each other and placed randomly in space, whereas the distance between their corresponding binding interfaces were under the range of a given distance cutoff *d_c_*. Following the initial conformation, both proteins made random diffusions within each simulation step. The probability of accepting the new configuration after diffusions was based on the calculation of the binding energy between two interacting proteins. The binding energy between two proteins during association was described by a simple physically based potential function that consisted of three terms: the first component was the electrostatic interaction, the second component was the hydrophobic effect, and the third component was associated with the exclusive volume effect during protein binding. More specifically, the simplified version of the electrostatic interaction was followed by the Kim–Hummer model [57,58], whereas the hydrophobic effect between a pair of contact residues were adopted from the Kyte and Doolittle model [59]. If the new conformation was accepted after each simulation step, we would further count how many native contacts were restored. On the basis of our previous study [27], when there were at least three native contacts restored, we assumed that the ligand and the receptor had successfully formed an encounter complex, so that the current simulation trajectory would be terminated. A native contact was considered as being restored if the distance between the representative centers of the two residues was less than 2Å from the distance observed in the native structure of the complex. In contrast, if an encounter complex cannot be formed through the predefined criteria, the simulation trajectory would continue until it reached the maximal time duration.

In practice, this simulation algorithm was performed under different distance cutoffs in order to estimate protein association effectively. Specifically, a large number (10^3^) of trajectories were generated for each value of distance cutoff. Each trajectory contained 10^3^ simulation steps, and the length of each simulation step was 1 ns. Therefore, the maximal time duration of each trajectory was 10^3^ ns long [27,28]. Among all these trajectories, the initial conformations were relatively different, but their initial distances of the binding interfaces between ligands and receptors were below the same cutoff value. As a result, ligand and receptor might diffuse away from each other within some of these trajectories, whereas encounter complexes could be found by the end of other trajectories. On the basis of counting the ratio of trajectories in which encounter complexes were successfully formed, we could plot the profile of association probability as a function of distance cutoff. In order to provide insights about the rate of association between ligands and receptors from different systems in our dataset, these calculated profiles were compared with each other. More details about this comparison are described in the Results section. Finally, it is worth mentioning that we applied a machine-learning-based algorithm in our previous study, in addition to the coarse-grained simulation, in order to predict the rates of association for a large benchmark set. The benchmark set in our previous study contained protein complexes with very diverse structural organizations. The purpose of machine learning was to account for the degrees of molecular flexibility embedding in these complexes. In contrast, the structures of all the ligand–receptor complexes in the dataset of this study were highly similar. Moreover, in this work, we tried to understand the mechanisms of binding kinetics between TNF ligands and receptors by only using simulations that were based on physical and chemical principles. Machine-learning algorithm, by its nature, does not belong to these categories. Although it was able to improve the prediction accuracy, it was not within the scope of our simulation analysis in this study. Therefore, the machine-learning algorithm was not considered in the current method.

### 4.3. The Brownian Dynamic Simulations of Ligand–Receptor Complex Dissociation

In order to further study the dissociation between a trimeric ligand and a monomeric receptor, another previously developed coarse-grained simulation method was applied to all the systems of TNF complexes in the dataset. This method was tested to study the dissociation between colicin E9 endonuclease (E9 DNase) and two immunity proteins, Im2 and Im9, as its binding partners. Our simulations were able to reflect the difference of binding specificity between E9 DNase/Im2 and E9 DNase/Im9 interactions [60]. Similar to the kinetic Monte Carlo simulation, the atomic structure of protein complexes in this method was also reduced by a simplified model, in which a residue was coarse-grained into two sites: one was the position of its Cα atom, whereas the other indicated the center of mass position for its sidechain. Starting from the native conformation, the position and the velocity of each residue in a complex was updated by the Brownian dynamic algorithm. Specifically, the change of position for each residue within each time step depended on its calculated velocity, whereas the change of velocity was further determined by the total external forces applied to the residue, including the forces from the interactions between residues and the stochastic force from surrounding environments. The calculation of the forces, on the other hand, was dependent on the positions of all residues in the system. As this process iterated, the overall conformation of the ligand–receptor complex evolved dynamically.

The interactions between residues consist of the short-range interactions along the backbone of residue, intra-molecular long-range interactions, and the inter-molecular interactions between ligands and receptors. The short-range interactions and intra-molecular long-range interactions are modeled by the traditional Go-like potential [61]. They are used to constrain the tertiary structure of each protein monomers in the complex around their native conformation, but still allow a small degree of flexibility. On the other hand, the strength of the inter-molecular interactions was derived from the statistics of available protein complexes in the structural database. As described in the previous study, we collected a large-scale structural library of protein complexes from the 3did database [62]. The library consists of 4960 entries of protein–protein interactions from either homodimers or heterodimers. The detailed energy parameters were derived on the basis of the corresponding numbers of interactions across the binding interfaces between a specific pair of residue types that are observed though all the complexes in the library. Consequently, we showed that these energy parameters can capture the first-principle physical and chemical characteristics of molecular recognition such as electrostatic interactions and hydrophobic effect. As a result, by incorporating both Go-like and knowledge-based potentials into the Brownian dynamic equation of motions, the ligand and receptor in the initial native complex would be able to dissociate from each other while maintaining their relative tertiary structures. In order to obtain statistically meaningful comparison of their dissociation processes, multiple (10) independent trajectories were carried out for all complexes in the dataset.

Finally, it is worth mentioning that the coarse-grained models in simulations of association and dissociation were slightly different. In the kinetic Monte Carlo simulation for protein association, the side-chain of each residue is represented by its functional center, depending on the physical-chemical property of each amino acid. This representation highlights the importance of electrostatic interactions in driving the process of association between two proteins when they are separated from each other. On the other hand, in the Brownian dynamic simulation for protein dissociation, the side-chain of each residue is simply represented by its center of mass. This is because the statistical potential used in the evaluation of inter-molecular interactions during dissociation was previously constructed on the basis of the representation of side-chain center of mass. In summary, during the development of these two simulation methods, the implementation of different coarse-grained representations were originated from the natures of different potentials and the features they aim to capture. Because the association or dissociation probabilities calculated from one method are independent to the other, the combinations of these probabilities to estimate the binding stability of protein complexes are not affected by the representation of each method.

## Figures and Tables

**Figure 1 ijms-21-01778-f001:**
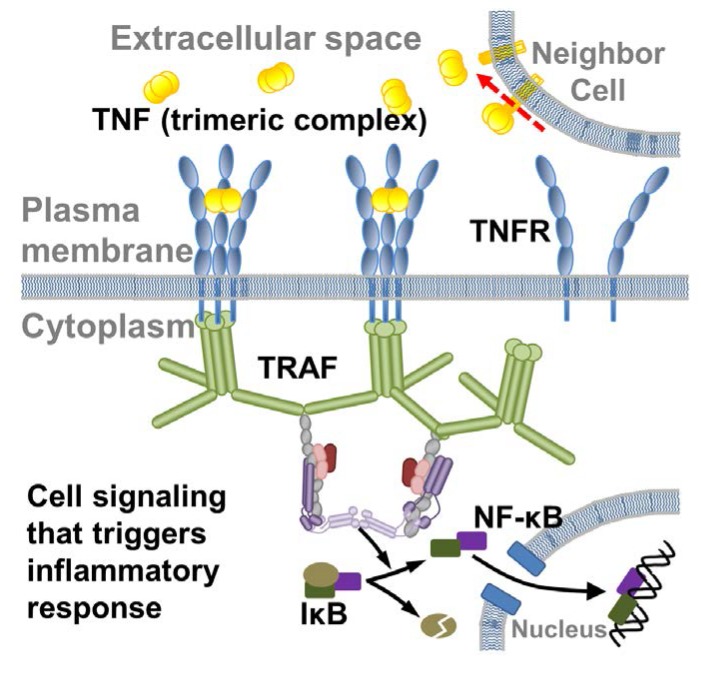
Ligands that belong to tumor necrosis factor (TNF) superfamily are originally expressed on cell surfaces. Most of them are oligomerized into trimers and later cleaved by metalloproteases. On cell plasma membrane, the binding between TNF ligands and their receptors in the tumor necrosis factor receptor (TNFR) superfamily initiates multiple intracellular signaling pathways. For instance, the cytoplasmic region of TNFR after ligand binding will recruit the scaffold protein called TNF receptor associated factor (TRAF), which will further turn on the NF-κB signaling pathway. NF-κB is an important transcription factor in inflammation as an activator of numerous pro-inflammatory cytokines and chemokines. The red dashed arrow in the figure indicates the cleavage of TNF ligands into their soluble forms, while the black arrows illustrate the downstream signaling pathways activated by TNF ligand-receptor interactions.

**Figure 2 ijms-21-01778-f002:**
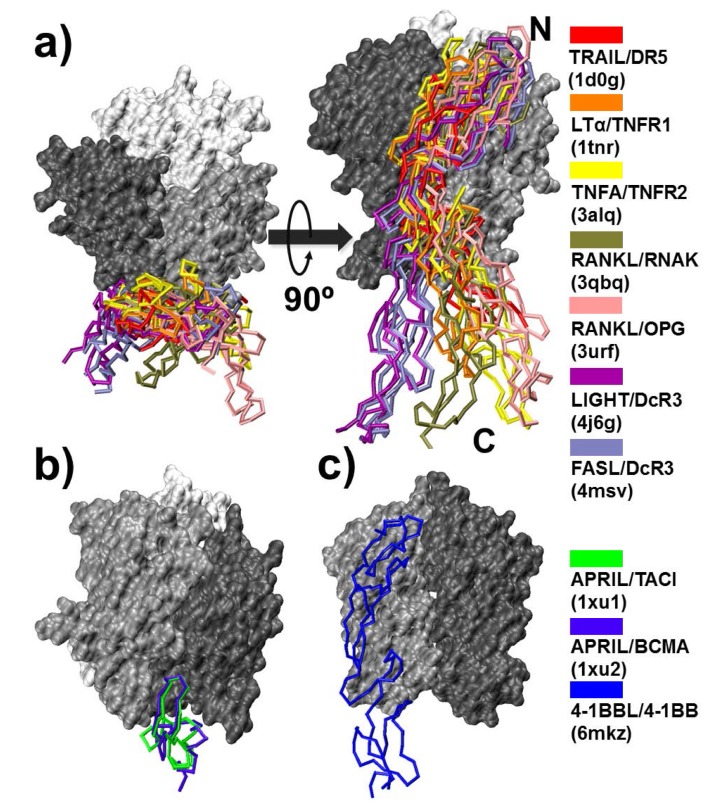
We compared the structure of the interactions between ligands and receptors of the 10 TNF/TNFR complexes in the dataset. The color index of these complexes is listed on the right side. Specficially, binding between ligands and receptors can be divided into three groups. Firstly, 7 out of 10 receptors in the dataset could be closely aligned with each other, as shown in (**a**). The second group, as shown in (**b**), is the structural comparison of a pair of functionally related receptors, B cell maturation antigen (BCMA; violet) and short form of transmembrane activator and CAML-interactor (TACI; green). Finally, the binding between receptor 4-1BB and its ligand 4-1BBL, as shown in (**c**), is highly different from the first two.

**Figure 3 ijms-21-01778-f003:**
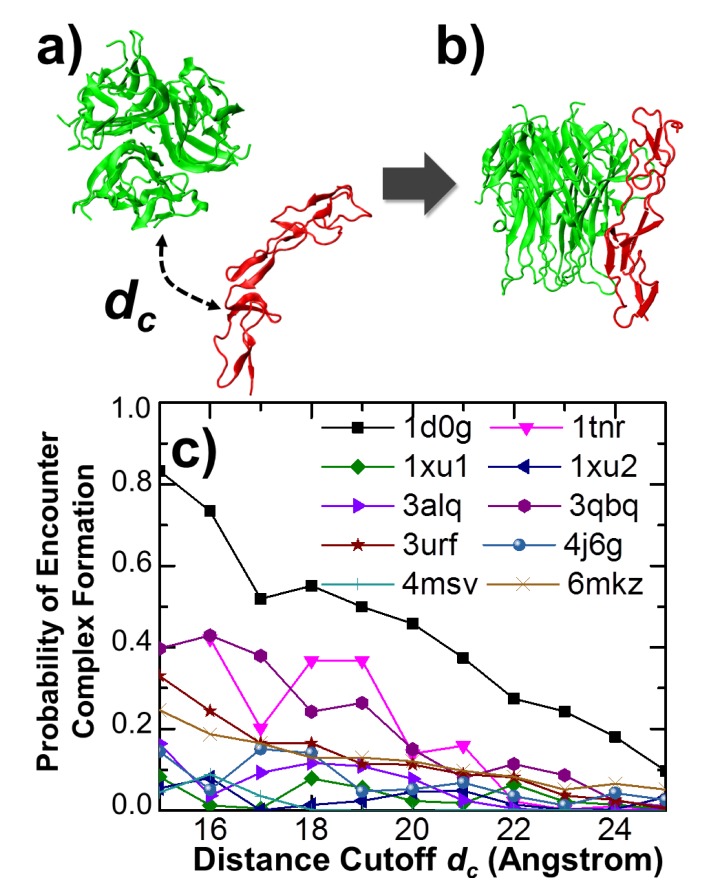
We applied a residue-based kinetic Monte Carlo method to simulate the association between ligands and receptors of all the 10 protein complexes. The monomeric receptor was first separated from its trimeric ligand within different values of distance cutoff (**a**). Under each distance cutoff, 10^3^ simulation trajectories were generated from different initial conformations. Ligands and receptors can successfully associate together within some of these trajectories (**b**). We then counted the probability of finding the encounter complexes among these trajectories. The association probabilities for different protein complexes are plotted in (**c**) as a function of distance cutoff.

**Figure 4 ijms-21-01778-f004:**
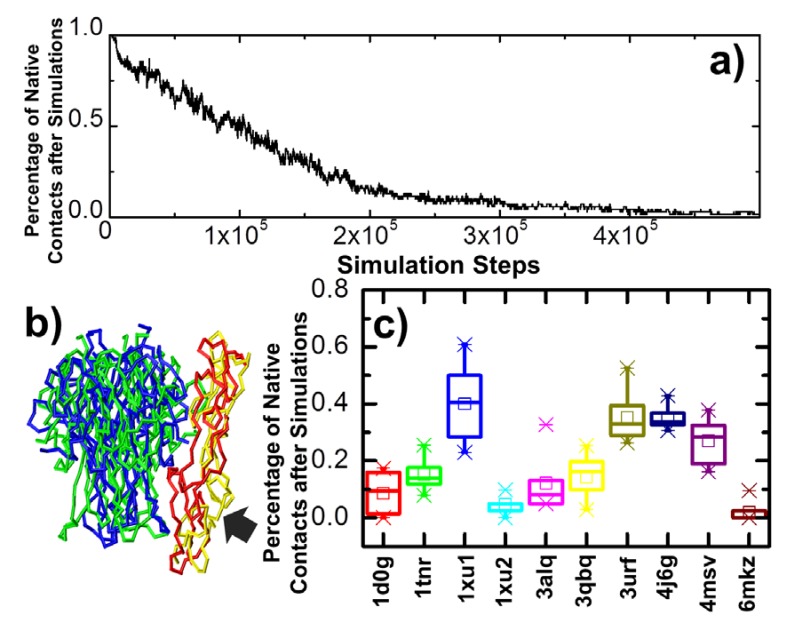
The coarse-grained Brownian dynamic simulations were carried out to compare the dissociation processes among all the 10 protein complexes in the dataset. We carried out 10 independent simulation trajectories for each complex from its native structural conformation. For each trajectory, we calculated the percentage of native contacts left in the dissociating complex along with the simulation time. The changes of this percentage in one of these trajectories is shown in (**a**) as a function of time for the complex LTα/TNFR1. The comparison between initial and final conformations of this specific trajectory is further shown in (**b**). The region with the largest structural variations in the receptor is highlighted by an arrow. Finally, the statistical distributions of native contact percentage remaining after simulations calculated from all 10 trajectories are plotted as a box-whisker diagram in (**c**) for all the complexes, wherein protein databank (PDB) IDs are indexed at the bottom. In the diagram, the average values of percentage are marked as small squares in the middle of each distribution, whereas the maximal and minimal values are marked as symbols of snowflakes. The box of each distribution in the plot indicates the range from 25 to 75 percentiles, whereas the whisker indicates the outlier of the distribution with the coefficient equal to 1.5.

**Figure 5 ijms-21-01778-f005:**
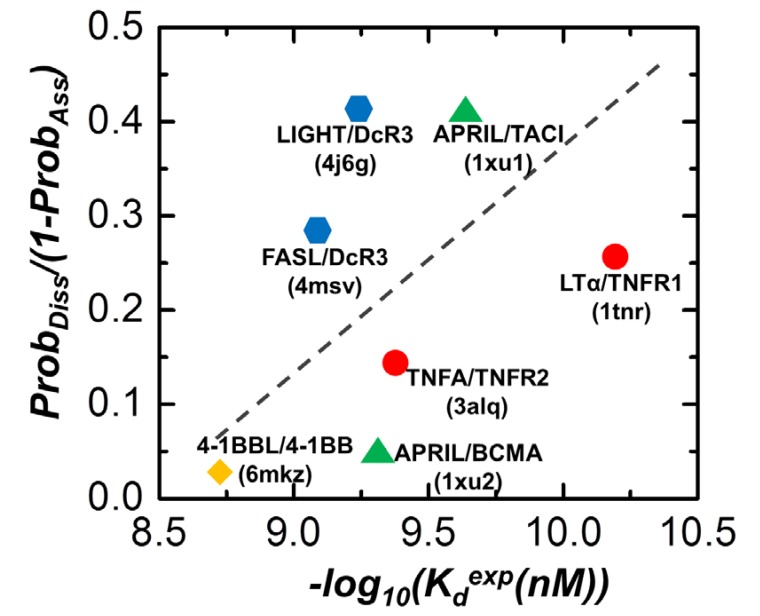
Among 10 protein complexes in the dataset, 7 of them had corresponding data of binding affinities available in the literature. In order to compare our simulation results with the available experimental measurements, the calculated probabilities of association and dissociation from our simulations were combined together and converted into a stability factor, which was positively correlated with the binding strength. As a result, the calculated stability factors for these seven complexes were directly correlated to the logarithm of their experimentally measured binding affinities. As shown by different symbols in the figure, the seven protein complexes were further divided into different subgroups, so that they could be compared within the specific biological context.

**Figure 6 ijms-21-01778-f006:**
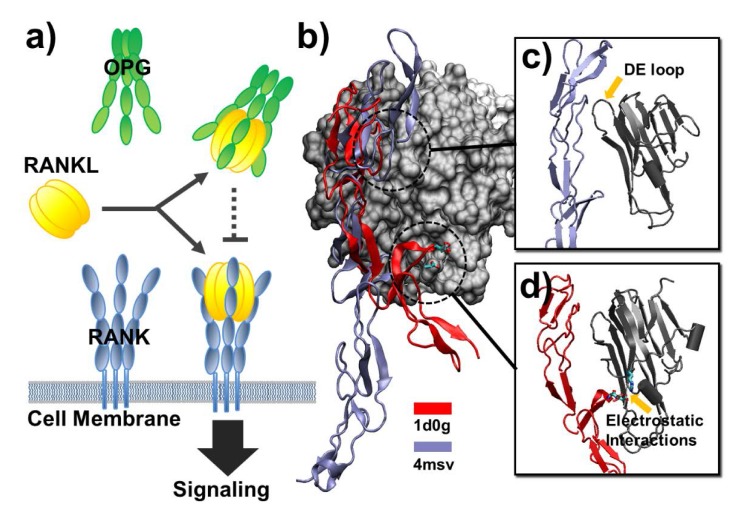
We provide insights to the biological functions of different ligand–receptor interactions. One example is the difference of binding between the ligand receptor activator of nuclear factor-κB ligand (RANKL) and its cell surface receptor receptor activator of nuclear factor-κB (RANK) versus its competitor, the soluble decoy receptor osteoprotegerin (OPG) (**a**). We found the slow dynamics of binding between ligand and decoy receptor. In order to explore the energetic origin of this phenomenon, we compared the ligand-binding structure of decoy receptor 3 (DcR3) with death receptor 5 (DR5) (**b**). The association and dissociation of decoy receptor DcR3 with its ligand were slow, whereas the binding and unbinding processes for DR5 were much faster. We suggest the extended hydrophobic interactions between the backbones of DcR3 and its ligand result in the slow binding kinetics and low selectivity (**c**). In contrast, electrostatic interactions between the polar residues at the interface of TNF-related apoptosis-inducing ligand (TRAIL)/DR5 complex led to fast binding kinetics and high binding specificity (**d**).

**Table 1 ijms-21-01778-t001:** The dataset of TNF ligands and receptors used in this study.

Index	Ligand	Receptor	Organism	PDB
1	TNFSF1 (LTα)	TNFR1	*Homo sapiens*	1tnr
2	TNFSF2 (TNFA)	TNFR2	*Homo sapiens*	3alq
3	APO2L/TRAIL	DR5	*Homo sapiens*	1d0g
4	RANKL	RANK	*Mus musculus*	3qbq
5	RANKL	OPG	*Homo sapiens*	3urf
6	LIGHT	DcR3	*Homo sapiens*	4j6g
7	FASL	DcR3	*Homo sapiens*	4msv
8	APRIL	TACI	*Homo sapiens*	1xu1
9	APRIL	BCMA	*Homo sapiens*	1xu2
10	4-1BBL	4-1BB	*Mus musculus*	6mkz

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
