# Peer review of "A Systematic Test of Receptor Binding Kinetics for Ligands in Tumor Necrosis Factor Superfamily by Computational Simulations"

_ijms, 2020, doi:10.3390/ijms21051778_

Round 1

Reviewer 1 Report

In this paper, the authors have proposed an interesting methodology to study the association and dissociation process of TNF/TNFR complexes. The methodology regroup coarse-grained simulations, one with Monte-Carlo Algorithm and one with Brownian Dynamics methodology. 10 different complexes have been studies and a comparison between simulation results and experimental data have been made.

The paper is interesting and very pleasant to read. However, several questions must be answered and details have to be added in the paper before publications in IJMS.

1) About the binding of the trimeric TNF with the receptor, figure 1 present a binding mode about a trimeric ligand with a trimeric TNFR. But the author have chosen to consider only a monomeric receptor. Why this choice ? Is the binding interface of the two other TNFR are symmetrical and identical to the one we observe in the X-ray srructure of the 10 complexes?

The question about this choice has a great importance in the simulations made. Do we know anything about the mechanism and the dynamics of those TNFR? Does the trimeric TNF receptor must assemble first before the binding of TNF or the trimeric ligand binds to a monomer before the other TNFRs bind to the TNF (as suggested by the simulations presented)? Is a trimeric TNFR is required for the triggering of the intracellular pathway involving TRAF?

Discussion about author's choice and the reliability of their model against what we know about the true molecular mechanism has to be added in the paper.

2) TNF receptors have a transmembrane segment. This segment has obivously a strong impact on the dynamics of the receptor, especially its diffusion. If i understood well, authors haven't considerd the membrane in their models. This limitation and the probable impact on their results have to be discussed in the article.

3) P16, authors have written: "If an encounter complex cannot be formed through the predefined criteria, the simulation trajectory will continue
until it reaches the maximal time duration." But we don't know how many steps of those simulations have been made. It is especially important because one of their conclusion is, p 7:" On the other hand, the low probabilities in many other systems suggest slow association between
ligands and receptors".

How the authors have verified that the step number of monte-carlo simulation is high enough to allow the association to occur ? What is the step number chosen ?

4) Still on association simulation, do the final structures are the same as the x-ray structures ? Do they retrieve any similar binding mode ? no words have been said about that in the current manuscript.

5) It is written p8 "The process of dissociation is guided by the interactions at their binding interfaces". Does it mean that Brownian dynamics simulations are biased? if yes, on which constraint ?

6) Authors have used a coarse-grained model, but it seems different between the monte-carlo simulation and the brownian dynamics. Indeed, they talk about a representative center and a center of mass (both for the representation of residues sidechains), respectively for MC simulations and BC simulations. Is it relevant then to combine the probabilities as they did in Figure 5? In my opinion, the authors must comment this aspect.

7) one last remark: I suggest to clarify the caption of Figure 4. Authors need to precise that 4a and 4b is only for one replicate of LTα/TNFR1 complex.

Author Response

Responses to Reviewer 1's Questions

******************************************************************************

In this paper, the authors have proposed an interesting methodology to study the association and dissociation process of TNF/TNFR complexes. The methodology regroup coarse-grained simulations, one with Monte-Carlo Algorithm and one with Brownian Dynamics methodology. 10 different complexes have been studies and a comparison between simulation results and experimental data have been made. The paper is interesting and very pleasant to read. However, several questions must be answered and details have to be added in the paper before publications in IJMS.

  • We really appreciate the positive comments from our reviewer!

1) About the binding of the trimeric TNF with the receptor, figure 1 present a binding mode about a trimeric ligand with a trimeric TNFR. But the author have chosen to consider only a monomeric receptor. Why this choice ? Is the binding interface of the two other TNFR are symmetrical and identical to the one we observe in the X-ray srructure of the 10 complexes?

The question about this choice has a great importance in the simulations made. Do we know anything about the mechanism and the dynamics of those TNFR? Does the trimeric TNF receptor must assemble first before the binding of TNF or the trimeric ligand binds to a monomer before the other TNFRs bind to the TNF (as suggested by the simulations presented)? Is a trimeric TNFR is required for the triggering of the intracellular pathway involving TRAF?

Discussion about author's choice and the reliability of their model against what we know about the true molecular mechanism has to be added in the paper.

  • We thank our reviewer for asking this interesting question. The binding interfaces of three TNFR to the TNF ligand are symmetric and identical as one observed in the X-ray structure of the 10 complexes. IN this study, we try to only focus on the energetic and structural features in binding of monomeric TNF receptor to its ligand in which the interface is located at groove regions between every two adjacent subunits of a trimer. Because before the study of collective behaviors which involve multiple ligands and receptors in their cellular environments, one needs to estimate the kinetic properties of binding between individual TNF and its monomeric receptor as a basis. Therefore, we choose to consider the binding between TNF ligand and only a monomeric receptor in this study. Moreover, TNF ligands generally organize into a homo-trimeric quaternary structure before cleaved by metalloproteases and exposed to TNFRs. After these trimeric ligands bind to their receptors, the ligand-receptor complexes can further organize into higher-order clusters, as observed in recent experiments for some member of TNFR superfamily. In summary, the discussions about our choice and the reliability of our model against what we know about the true molecular mechanism have been added into the revised manuscript. Please see page 14 for details.

2) TNF receptors have a transmembrane segment. This segment has obivously a strong impact on the dynamics of the receptor, especially its diffusion. If i understood well, authors haven't considerd the membrane in their models. This limitation and the probable impact on their results have to be discussed in the article.

  • We thank you our reviewer for the constructive suggestion. We have now mentioned this detail about the membrane confinement of TNF receptor in the context of our computational modeling study. As can be found in page 14 of revised manuscript, “TNF receptors are membrane proteins anchored on cell surfaces. Intuitively, binding of these membrane receptors to their ligands has very different properties from interactions of proteins in solution. One major reason is that the diffusions of TNFR are confined on cell surfaces. Unlike freely diffusive soluble proteins which possess of three translational and rotational degrees of freedom, membrane-anchored proteins not only lose one translational degree of freedom that is perpendicular to the membrane surface, but also obtain constraints on their rotational degrees of freedom. Moreover, TNF ligands generally organize into a homo-trimeric quaternary structure before cleaved by metalloproteases and exposed to TNFRs. This leads into the assembly of a minimal ligand-receptor functional unit with a 3:3 stoichiometric combination. It has been further found that, for different members in TNFRSF, the formation of these complexes can trigger their spatial oligomerization into higher-order clusters and initiate intracellular signaling pathways. These dynamic processes can be simulated by a lower-resolution computational model recently developed by our lab. However, before the study of collective behaviors which involve multiple ligands and receptors in their cellular environments, one needs to estimate the kinetic properties of binding between individual TNF and its monomeric receptor as a basis. The computational mothed in this paper can isolate the effect of ligand-receptor binding from a number of above-mentioned complexity on the cellular level. As a result, we will be able to only focus on the energetic and structural features in binding of monomeric TNF receptor to the ligand in which the interface is located at groove regions between every two adjacent subunits of a trimer. In the future, the association and dissociation rates derived from this study for individual pair of ligand-receptor interactions can then be fed into our lower-resolution simulation method to understand the mechanism of receptor clustering on cell surfaces as a multiscale modeling framework”.

3) P16, authors have written: "If an encounter complex cannot be formed through the predefined criteria, the simulation trajectory will continue until it reaches the maximal time duration." But we don't know how many steps of those simulations have been made. It is especially important because one of their conclusion is, p 7:" On the other hand, the low probabilities in many other systems suggest slow association between ligands and receptors".

How the authors have verified that the step number of monte-carlo simulation is high enough to allow the association to occur ? What is the step number chosen ?

  • We are sorry for not mentioning this technical detail. To estimate the probability of association, a large number of kinetic Monte-Carlo simulation trajectories were carried out, and we then counted the ratio that encounter complexes were successfully formed among all these trajectories. In specific, each trajectory contains 103 simulation steps, and the length of each simulation step is 1ns. Therefore, the maximal time duration of each trajectory is 103ns long. The values of these parameters are adopted from our previous studies which have been tested against experimental benchmark set. Please see page 17 in the revised manuscript for details. We appreciate the efforts that our reviewer put to make the paper better.

4) Still on association simulation, do the final structures are the same as the x-ray structures ? Do they retrieve any similar binding mode ? no words have been said about that in the current manuscript.

  • We thank our reviewer for mentioning this issue. We are sorry for being inconsiderate. We have now added the detailed criteria of how to judge if an encounter complex has formed along the simulation of association. In specific, as shown in page 17 of the revised manuscript, “if the new conformation is accepted after each simulation step, we will further count how many native contacts are restored. Based on our precious study, when there are at least three native contacts restored, we assumed that the ligand and the receptor have successfully formed an encounter complex, so that the current simulation trajectory will be terminated. A native contact is considered to be restored if the distance between the representative centers of the two residues is less than 2Å from the distance observed in the native structure of the complex. In contrast, if an encounter complex cannot be formed through the predefined criteria, the simulation trajectory will continue until it reaches the maximal time duration”.

5) It is written p8 "The process of dissociation is guided by the interactions at their binding interfaces". Does it mean that Brownian dynamics simulations are biased? if yes, on which constraint ?

  • We thank our reviewer for asking this question. The Brownian dynamic simulations are not biased and there is no constraint to drive the process of dissociation in the simulations. The word “guided” in the original manuscript might be misleading. We are sorry for not being precise. The information we tried to deliver is that the process of dissociation is the consequence of gradually losing intermolecular interaction at the binding interface of the complex, when ligand and receptor are fluctuating around their native conformation after simulation. We have now changed the sentence into following: “the intermolecular interactions formed in the initial native conformation by residues between ligands and receptors gradually break under a stochastic background in the Brownian dynamic simulations, which lead into the final dissociation of the complex. The parameters of these intermolecular interactions between different types of residues were derived from a knowledge-based potential, as introduced in the methods”. Please see page 8 in the revised manuscript for details.

6) Authors have used a coarse-grained model, but it seems different between the monte-carlo simulation and the brownian dynamics. Indeed, they talk about a representative center and a center of mass (both for the representation of residues sidechains), respectively for MC simulations and BC simulations. Is it relevant then to combine the probabilities as they did in Figure 5? In my opinion, the authors must comment this aspect.

  • We thank our reviewer for mentioning this issue. We applied kinetic Monte-Carlo simulations to study the association between ligands and receptor, and applied Brown dynamic simulation to study their dissociation. In both methods, coarse-grained representations of protein structures were used. However, two coarse-grained models are slightly different. In the kinetic Monte-Carlo simulation, the side-chain of each residue is represented by its functional center, depending on the physical-chemical property of each amino acid. This representation highlights the importance of electrostatic interactions in driving the process of association between two proteins when they are separated from each other. On the other hand, in the Brownian dynamic simulation, the side-chain of each residue is simply represented by its center of mass. This is because the statistical potential used in the evaluation of inter-molecular interactions during dissociation was previously constructed based on the representation of side-chain center of mass. In summary, during the development of these two simulation methods, the implementation of different coarse-grained representations were originated from the natures of different potentials and the features they aim to capture. Finally, because the association or dissociation probabilities calculated from one method are independent to the other, the combinations of these probabilities to estimate the binding stability of protein complexes are not affected by the representation of each method. We have now added above clarification into the revised manuscript after we introduced the method of Brownian dynamic simulation. Please see page 19 for details.

7) one last remark: I suggest to clarify the caption of Figure 4. Authors need to precise that 4a and 4b is only for one replicate of LTα/TNFR1 complex.

  • We are sorry for not being precise. We have now added the clarification in the caption of Figure 4 as follows: “(a) shows the changes of this percentage as a function of time in one of these trajectories for the complex LTα/TNFR1. The comparison between initial and final conformations of this specific trajectory is further shown in (b).” Please see page 26 in the revised manuscript for details. We appreciate the efforts that our reviewer put to make the paper better.

Reviewer 2 Report

This article represents a reasonably rigorous attempt to computationally quantify the different effects governing physiologically relevant associative dynamics of TNF-related proteins. Using separate protocols, it accounts for both associative and dissociative kinetics, as well as binding energy. The scope of work, and the specific application, are sufficiently novel as to warrant publication.

One source of puzzlement is the fact that, in 2017, this same research group examined binding associative kinetics using coarse grained simulations with machine learning employed to enhance perception of molecular flexibility effects, yet in this paper they left out the learned flexibility. Why? Did they not feel they had adequate training data?  Did the ML correction detract from simpler rigid-body analysis?

One other criticism is the paper seems to pay inadequate attention to recent work done in quantifying molecular dissociation via Brownian dynamics. Some papers that might be of relevance include:
- Hollingsworth SA, Nguyen BD, Chreifi G, Arce AP, Poulos TL. Insights into the Dynamics and Dissociation Mechanism of a Protein Redox Complex Using Molecular Dynamics. J Chem Inf Model. 2017;57(9):2344–2350.
- Pan AC, Jacobson D, Yatsenko K, Sritharan D, Weinreich TM, Shaw DE. Atomic-level characterization of protein-protein association. Proc Natl Acad Sci U S A. 2019;116(10):4244–4249.

There are lots of other Brownian papers to consider as well -- it seems to be a hot topic right now.

An additional minor point is that the interpretability of Figs. 2-4 would benefit from having the gene/protein name listed, rather than the PDB ID.

Beyond that, while the paper is fairly well written, there are a number of grammatical ambiguities that distract from the work and, at times, confuse the message. The paper would benefit from a bit more careful proofreading.

Author Response

Responses to Reviewer 2's Questions

******************************************************************************

This article represents a reasonably rigorous attempt to computationally quantify the different effects governing physiologically relevant associative dynamics of TNF-related proteins. Using separate protocols, it accounts for both associative and dissociative kinetics, as well as binding energy. The scope of work, and the specific application, are sufficiently novel as to warrant publication.

  • We really appreciate the positive comments from our reviewer!

One source of puzzlement is the fact that, in 2017, this same research group examined binding associative kinetics using coarse grained simulations with machine learning employed to enhance perception of molecular flexibility effects, yet in this paper they left out the learned flexibility. Why? Did they not feel they had adequate training data?  Did the ML correction detract from simpler rigid-body analysis?

  • We thank our reviewer for mentioning this interesting point. We have applied a machine-learning based algorithm in our previous study, in addition to the coarse-grained simulation, to predict the rates of association for a large benchmark set. The benchmark set in our previous study contains protein complexes with very diverse structural organizations. The purpose of machine learning is to account for the degrees of molecular flexibility embedding in these complexes. In contrast, the structures of all the ligand-receptor complexes in the dataset of this study are highly similar. Moreover, in this work, we try to understand the mechanisms of binding kinetics between TNF ligands and receptors by only using simulations that are based on physical and chemical principles. Machine-learning algorithm, by its nature, does not belong to these categories. Although it was able to improve the prediction accuracy, it is not within the scope of our simulation analysis in this study. Therefore, the machine-learning algorithm is not considered in current method. We have now added above clarification into the revised manuscript when we introduced the method of kinetic Monte-Carlo simulation. Please see page 17 for details.

One other criticism is the paper seems to pay inadequate attention to recent work done in quantifying molecular dissociation via Brownian dynamics. Some papers that might be of relevance include:

- Hollingsworth SA, Nguyen BD, Chreifi G, Arce AP, Poulos TL. Insights into the Dynamics and Dissociation Mechanism of a Protein Redox Complex Using Molecular Dynamics. J Chem Inf Model. 2017;57(9):2344–2350.

- Pan AC, Jacobson D, Yatsenko K, Sritharan D, Weinreich TM, Shaw DE. Atomic-level characterization of protein-protein association. Proc Natl Acad Sci U S A. 2019;116(10):4244–4249.

There are lots of other Brownian papers to consider as well -- it seems to be a hot topic right now.

  • We thank our reviewer for the information. We are sorry for not paying adequate attention to recent work done in quantifying protein-protein association and dissociation via molecular dynamic simulations. We have now added the citations recommended by our reviewer into the revised manuscript, as well as some other relevant references on understanding protein interaction by molecular dynamic simulations. In specific, these statements and citations have been included into the Introduction section as following: “All-atom MD simulations have recently been utilized to understand the molecular mechanisms during the processes of protein-protein association [JCIM, 2017], as well as dissociation [PNAS, 2019]. Unfortunately, due to the high demands for computational resources, this technique has so far only been successfully applied to a limited number of protein complexes [other related references]”. Please see page 4 in the revised manuscript for details.

An additional minor point is that the interpretability of Figs. 2-4 would benefit from having the gene/protein name listed, rather than the PDB ID.

  • We thank our reviewer for the constructive suggestion. We have now added protein name into the list of Figure 2 to give a better way of the correspondence between PDB ID and names of these protein complexes in the figure. Please see the new Figure 2 in the revised manuscript. We appreciate the efforts that our reviewer has put to make the paper better.

Beyond that, while the paper is fairly well written, there are a number of grammatical ambiguities that distract from the work and, at times, confuse the message. The paper would benefit from a bit more careful proofreading.

  • We thank our reviewer for the suggestion. We have now carefully proofread the revised manuscript and correct the grammatical errors, as well as the confusing messages in the text. We really appreciated the efforts our reviewer put to make the paper better.

Round 2

Reviewer 1 Report

The paper has been greatly improved by the authors. Their new comments are relevant, clear and strengthen the article. It is now acceptable for publication in IJMS.

I suggest the authors to carrefully read once again the manuscript, since some typos are still present (for example, p17 "Based on our precious study"; I assumed that it is "previous", not "precious").

Author Response

The paper has been greatly improved by the authors. Their new comments are relevant, clear and strengthen the article. It is now acceptable for publication in IJMS.

I suggest the authors to carrefully read once again the manuscript, since some typos are still present (for example, p17 "Based on our precious study"; I assumed that it is "previous", not "precious").

Answer: We thank our reviewer for taking time to proofread the revised manuscript. We have now carefully read the manuscript once again and have corrected all the typos we can find. Again, we appreciate our reviewer for making our paper better!